# First-Line Treatment for Primary Breast Diffuse Large B-Cell Lymphoma Using Immunochemotherapy and Central Nervous System Prophylaxis: A Multicenter Phase 2 Trial

**DOI:** 10.3390/cancers12082192

**Published:** 2020-08-06

**Authors:** Ho-Young Yhim, Dok Hyun Yoon, Seok Jin Kim, Deok-Hwan Yang, Hyeon-Seok Eom, Kyoung Ha Kim, Yong Park, Jin Seok Kim, Hyo Jung Kim, Cheolwon Suh, Won Seog Kim, Jae-Yong Kwak

**Affiliations:** 1Department of Internal Medicine, Jeonbuk National University Medical School, Jeonju 54907, Korea; yhimhy@jbnu.ac.kr; 2Department of Oncology, Asan Medical Center, University of Ulsan College of Medicine, Seoul 05505, Korea; dhyoon@amc.seoul.kr (D.H.Y.); csuh@amc.seoul.kr (C.S.); 3Department of Medicine, Samsung Medical Center, Sungkyunkwan University School of Medicine, Seoul 06351, Korea; kstwoh@skku.edu; 4Department of Internal Medicine, Chonnam National University Hwasun Hospital, Jeollanam-do 58128, Korea; drydh1685@hotmail.com; 5Center for Hematologic Malignancy, National Cancer Center, Goyang 10408, Korea; hseom@ncc.re.kr; 6Department of Internal Medicine, Soonchunhyang University College of Medicine, Seoul 04401, Korea; kyoungha@schmc.ac.kr; 7Department of Internal Medicine, Anam Hospital, Korea University College of Medicine, Seoul 02841, Korea; paark76@hanmail.net; 8Department of Internal Medicine, Severance Hospital, Yonsei University College of Medicine, Seoul 03722, Korea; hemakim@yuhs.ac; 9Department of Internal Medicine, Hallym University Sacred Heart Hospital, Anyang 14068, Korea; hemonc@hanmail.net

**Keywords:** central nervous system, diffuse large B-cell lymphoma, primary breast lymphoma, prophylaxis, rituximab

## Abstract

There are limited data from prospective controlled trials regarding optimal treatment strategies in patients with primary breast diffuse large B-cell lymphoma (DLBCL). In this phase 2 study (NCT01448096), we examined the efficacy and safety of standard immunochemotherapy and central nervous system (CNS) prophylaxis using intrathecal methotrexate (IT-MTX). Thirty-three patients with newly diagnosed primary breast DLBCL received six cycles of rituximab plus cyclophosphamide, doxorubicin, vincristine, and prednisone (R-CHOP) and four fixed doses of IT-MTX (12 mg). The median age was 50 years (range, 29–75), and all patients were females. According to the CNS-International Prognostic Index, most patients (*n* = 28) were categorized as the low-risk group. Among the 33 patients, 32 completed R-CHOP, and 31 completed IT-MTX as planned. With a median follow-up of 46.1 months (interquartile range (IQR), 31.1–66.8), the 2-year progression-free and overall survival rates were 81.3% and 93.5%, respectively. Six patients experienced treatment failures, which included the CNS in four patients (two parenchyma and two leptomeninges) and breast in two patients (one ipsilateral and one contralateral). The 2-year cumulative incidence of CNS relapse was 12.5%. Although standard R-CHOP and IT-MTX without routine radiotherapy show clinically meaningful survival outcomes, this strategy may not be optimal for reducing CNS relapse and warrants further investigation.

## 1. Introduction

Primary breast diffuse large B-cell lymphoma (DLBCL) is a rare subtype of primary extranodal non-Hodgkin lymphoma (NHL), consisting of approximately 1% of all NHL [1]. Most patients with primary breast DLBCL have localized-stage disease at diagnosis, but their outcomes are quite poor [2,3,4,5,6,7,8]. Although it is not clear whether primary breast DLBCL differs intrinsically from nodal DLBCL, it has different clinical features from nodal DLBCL, which shows predominant relapse in specific extranodal organs, that is, breast and central nervous system (CNS) [2,3,4,5,6,7,8]. Particularly, relapse in the CNS is a major issue because of its dismal prognosis. However, because most data on CNS relapse in primary breast DLBCL are largely based on retrospective reports with a small-to-moderate number of patients [2,3,4,5,6,7], the risk of CNS relapse and need for routine use of CNS prophylaxis have never been prospectively evaluated and remain debated.

Several retrospective studies have suggested that the use of consolidative radiotherapy to primary lesions could reduce loco-regional failure [5,8]. However, the impact of its use on survival outcomes is still controversial because the survival benefit of consolidative radiotherapy was predominantly observed in studies in the pre-rituximab era [9], suggesting that adding loco-regional control by radiotherapy may help overcome the inferior systemic therapy of the pre-rituximab era. Moreover, no studies have investigated long-term adverse effects and the risk of secondary malignancies associated with the use of radiotherapy. As breast cancer is one of the most common malignancies in females worldwide [10], minimization of the risk is of paramount importance in survivors. Thus, although the rituximab plus cyclophosphamide, doxorubicin, vincristine, and prednisone (R-CHOP) therapy has been established as the standard first-line therapy in newly diagnosed nodal DLBCL patients [11], there is no consensus on the optimal treatment strategy because of the paucity of controlled prospective data. Therefore, we designed a standard R-CHOP immunochemotherapy and CNS prophylaxis using intrathecal methotrexate (IT-MTX) to examine the improvement in progression-free survival (PFS) outcome by reducing CNS relapse. In addition, considering the improvement of lymphoma control by R-CHOP immunochemotherapy and minimization of the risk of secondary breast cancer, we omitted routine consolidative radiotherapy.

We report here the results of our prospective trial in patients with primary breast DLBCL, which addressed the efficacy and safety of R-CHOP plus CNS prophylaxis using IT-MTX without routine consolidative radiotherapy.

## 2. Results

### 2.1. Patient Characteristics

From January 2012 to July 2017, 33 female patients were enrolled and received study treatment, all of whom were included in the primary analyses (Figure 1). Baseline characteristics were described in Table 1. The median age was 50 years (range, 29–75). All but one patient had unilateral breast lymphoma, and nodal involvement was observed in 16 patients, primarily in the regional nodes (14 patients). Two patients had distant nodal involvement in each of the retroperitoneal and inguinal nodes. Thus, the Ann Arbor stage was IE in 17 patients, IIE in 13 patients, IIIE in two patients, and IVE in one patient, who had bilateral breast DLBCL with regional nodal involvement. None of patients had baseline CNS involvement. According to the CNS-International Prognostic Index (CNS-IPI) score, most (28 patients) were categorized as having low risk for developing CNS relapse, but no patient was placed in the high-risk group.

### 2.2. Feasibility and Treatment Response

Thirty-two of the 33 patients completed the R-CHOP treatment as pre-determined, as one patient withdrew a consent after four cycles of R-CHOP because of concerns about tolerance (Figure 1). Thirty-one patients completed CNS prophylaxis using IT-MTX as pre-determined, but two patients prematurely discontinued it because of refusal. These two patients received two and three doses of IT-MTX, respectively. Thus, 30 patients (90%) completed the study treatment as planned. One patient with bulky disease, who had suspected remnant lesion on the end-of-treatment positron emission tomography (PET) scan, received consolidative radiotherapy.

Thirty-two patients were evaluable for treatment response at the end of study treatment, and all these patients achieved complete response (CR).

### 2.3. Survival, Patterns of Relapse, and CNS Outcomes

At data cutoff (10 July 2019), all patients who were in the active follow-up phase had at least 2 years of follow-up. With a median follow-up of 46.1 months (interquartile range (IQR), 31.1–66.8), 6 patients had relapsed DLBCL and three of these died. Thus, the 2-year PFS and overall survival (OS) rates were 81.3% (95% confidence interval (CI), 67.7–94.8) and 93.5% (95% CI, 84.9–100.0), respectively (Figure 2A,B).

Of the six patients with lymphoma relapse, the sites of relapse were predominantly extranodal organs, that is, CNS in four patients and breast in two patients. Among the four patients with CNS relapse, three had isolated CNS relapse (brain parenchyma in two patients and leptomeninges in one patient) and one had concomitant meningeal and systemic nodal relapse. All four patients with CNS relapse had completed the study treatment as planned. Thus, the 2-year cumulative incidence of CNS relapse was 12.5% (95% CI, 0.3–23.2; Figure 2C). According to CNS-IPI, the 2-year cumulative incidence of CNS relapse in patients with low CNS-IPI risk was 10.7% (95% CI, 0–21.5), which was not statistically different from that observed in those with intermediate risk (25.0%; 95% CI, 0–57.4; *p* = 0.379), although the number of patients with intermediate risk was small (Figure 2D). Other baseline clinical variables were not significantly associated with CNS relapse as well (Appendix A). All four patients with CNS relapse received salvage treatment with high-dose MTX-based chemotherapy ± systemic chemotherapy, but three patients eventually died because of CNS (*n* = 1) or systemic (*n* = 2) progression. Only one patient had undergone salvage autologous transplantation after high-dose MTX-based salvage treatment and was disease-free at the time of final data collection.

Two patients with breast relapse, ipsilateral (*n* = 1) and contralateral (*n* = 1), received salvage systemic chemotherapy, followed by consolidative radiotherapy and autologous transplantation, respectively. These two patients were alive in remission.

### 2.4. Safety and Long-Term Survivorship

Toxicity was assessed in all 196 cycles of 33 patients (Table 2). The most common adverse event was hematological toxicity. Grade ≥ 3 neutropenia and thrombocytopenia occurred during 68 (34.7%) and 8 (4.1%) cycles, respectively. Febrile neutropenia occurred in 8 (4.1%) cycles was resolved by appropriate antibiotic treatment and supportive care. One patient had severe headache with temporary loss of consciousness after IT-MTX, which soon resolved spontaneously. No treatment-related deaths were observed.

At a median interval of 36.9 months (IQR, 24.5–59.9) from the last R-CHOP treatment, breast cancer occurred in four survivors (Table 3). Thus, the 3-year cumulative incidence of subsequent breast cancer was 10.2% (95% CI, 2.5–24.6; Appendix A).

## 3. Discussion

To our knowledge, this is the first prospective, multicenter trial on R-CHOP with overall CNS prophylaxis in primary breast DLBCL. In 33 primary breast DLBCL patients, all of whom were treated with R-CHOP plus IT-MTX without routine radiotherapy, we observed a clinically meaningful PFS (2-year, 81.3%) and OS (2-year, 93.5%), although the study did not reach its primary objective.

Consistent with previous retrospective studies [2,3,4,5,6,7,8], we observed high proportion of extranodal relapses, particularly in the CNS. In our previous retrospective report [2], we observed a 2-year CNS relapse rate of 16.3%, the majority of whom were treated with R-CHOP alone. However, despite overall prophylaxis using IT-MTX, we observed still high rate of CNS relapse (12.5% at 2 years) in this phase 2 trial, suggesting only a modest effect of IT-MTX on preventing CNS relapse in patients with primary breast DLBCL. Indeed, although the addition of rituximab to systemic chemotherapy has slightly reduced CNS relapse in DLBCL through improved systemic control [12], there are some data showing that CNS prophylaxis using IT-MTX or liposomal cytarabine ± hydrocortisone is not enough to prevent CNS relapse in nodal and extranodal forms of DLBCL [4,12,13,14,15,16]. Whereas some data favored the use of systemic infusion of high-dose MTX for CNS prophylaxis [17,18]. However, this approach might be associated with considerable toxicity requiring a careful assessment of risk/benefit ratio. Given that CNS relapse eventually occurred in approximately 10–20% of patients with primary breast DLBCL [2,3,4,8], most patients would have to face the unnecessary risk of treatment-related toxicities if high-dose MTX was universally infused to all primary breast DLBCL patients. Several primary breast DLBCL series have suggested that some clinical variables such as presence of regional nodal disease [3], stage-modified IPI [3], and tumor > 5 cm [6] might be associated with increased risk of CNS relapse, but these findings were not consistent across the studies. Furthermore, CNS-IPI was not effective for stratifying patients according to the risk of CNS relapse in our study. One possible step in solving this problem may be to investigate the biological features of primary breast DLBCL. DLBCL associated with translocation of *MYC* and *BCL2* and/or *BCL6* (double-hit or triple-hit lymphomas) needs to be investigated in the context of high risk for CNS relapse of primary breast DLBCL [19]. Recent data applying genetic analysis also revealed the molecularly defined high-grade group, which was characterized by similar molecular features and clinical outcomes of double-hit or triple-hit lymphomas [20,21]. Moreover, DLBCL with dual expression of the MYC and BCL2 proteins (double expresser DLBCL) had a poor prognosis with increased risk of CNS relapse, predominantly associated with non-germinal center B-cell (non-GCB) subtype, as per Hans algorithm [22]. Previous data have shown that primary breast DLBCL is closely associated with non-GCB subtype [2,23,24], suggesting the prognostic impact of double expresser status on CNS outcomes in primary breast DLBCL. Therefore, further studies are needed to explore the biological features that affect the risk of CNS relapse in patients with primary breast DLBCL. These biologic features may affect the selection of CNS prophylaxis in primary breast DLBCL patients. Given the recent retrospective data that appropriately selected limited-stage DLBCL (i.e., completely resected or interim PET-negative lymphomas) might be successfully treated with short-course immunochemotherapy without radiotherapy [25,26], a large subgroup of patients who did not have any biologic features for poor clinical outcomes might have excellent outcomes with a low risk for CNS failure. These patients may be eligible for study with de-escalating treatment strategies. Alternatively, patients with distinct biologic features for CNS failure may be addressed to find optimal prophylactic strategies incorporating high-dose MTX or other novel agents for preventing CNS relapse. Notably, ibrutinib and lenalidomide have been known to be active for CNS DLBCL [27,28], and both drugs have recently shown the feasibility in combination with R-CHOP [29,30]. Thus, it will be interesting to evaluate the impact of these drugs on the prevention of CNS relapse in primary breast DLBCL patients with adverse biologic features.

Another issue in the treatment of primary breast DLBCL is the role of radiotherapy. We evaluated the routine radiotherapy-free strategy; radiotherapy was only permitted if patients had bulky disease or remnant lesion on end-of-treatment PET scan. This is because previous reports showed conflicting results about the role of radiotherapy in terms of survival outcomes, even though it consistently improved local control [2,3,4,8]. In this study, relapse in the breast had occurred in two patients, but ipsilateral breast relapse had occurred in only one patient, who received salvage chemotherapy followed by involved-field radiotherapy and showed prolonged survival in remission. Furthermore, in the previous series, approximately 30–67% of relapses in the breast occurred in the contralateral breast [2,3,5], which would not have been prevented by radiotherapy to the primary lesion because it was regarded a consequence of systemic disease progression. Thus, it would lead to overtreatment if radiotherapy was delivered to all patients with primary breast DLBCL solely for improving local control. Interestingly, it is noteworthy that the risk of subsequent breast cancer in survivors was not negligible (3-year, 10.2%) in our study. As primary breast DLBCL is more prevalent in young- and middle-aged females, radiotherapy itself may subject them to more exaggerated risk of subsequent breast cancer, even though it varies according to age at presentation, radiation dose, and field size. Given the substantial risk of subsequent breast cancer and the low rate of ipsilateral breast relapse after R-CHOP immunochemotherapy, the need for investigations on radiation-free strategies is now apparent. However, it must also be noted that the current trial was not powered to directly determine whether consolidative radiotherapy had a definitive benefit for patients with primary breast DLBCL. Nevertheless, the results from this study suggest consolidative radiotherapy might be omitted in primary breast DLBCL patients, who achieved CR on end-of-treatment PET scan after first-line R-CHOP immunochemotherapy.

Our study has several limitations. First, our study is a phase 2 trial that included a rather small number of patients with a lack of randomized design. Although the survival benefit in a randomized, phase 3 trial provides a definitive conclusion, it is not feasible to conduct a randomized trial because of the rarity of this subtype of primary extranodal DLBCL. Moreover, the lack of prior prospective data on primary breast DLBCL compelled us to use the small number of historical control as a reference, which was retrospective data in nature. Therefore, the results of our study should be interpreted with caution, considering this limitation. Second, most CNS relapse occurred within first 2 years of initial diagnosis in nodal DLBCL [12], whereas late CNS relapse up to 10 years were suggested in primary breast DLBCL [4]. Thus, our cohort should be observed additionally to monitor late CNS relapse. Nevertheless, our study is the first to prospectively evaluate the treatment outcomes of standard R-CHOP therapy with CNS prophylaxis in patients with primary breast DLBCL.

## 4. Materials and Methods

### 4.1. Study Design and Patients

We carried out a prospective multicenter single-arm phase 2 study on R-CHOP in combination with prophylactic IT-MTX as a primary therapy in patients with newly diagnosed primary breast DLBCL. The Consortium for Improving Survival of Lymphoma (CISL) study group ran the study in nine academic institutions in South Korea. Patients were eligible if they were female; aged between 20 and 75 years; and had histologically confirmed, primary breast DLBCL without any previous treatment history for DLBCL. Primary breast lymphoma was defined as lymphoma involving only one or both breasts as a major extranodal site regardless of specific nodal involvement status [7]. Patients were excluded if they had Eastern Cooperative Oncology Group performance status > 2; inadequate hematological (absolute neutrophil count < 1500/μL or platelet < 75,000/μL), hepatic (aspartate or alanine aminotransferase > 3 times the upper limit of normal (ULN), total bilirubin >2 times the ULN), renal (serum creatinine ≥ 2 mg/dL), and cardiac (left ventricular ejection fraction < 50%) functions; malignancy within the last 5 years; human immunodeficiency virus (HIV) or hepatitis C seropositivity; and were pregnant or lactating. Patients who had DLBCL with widespread multiple extranodal organ involvements as well as breast were also excluded.

Pretreatment evaluation included complete blood count and differential; serum biochemistry with lactate dehydrogenase; serology tests for HIV and hepatitis B and C; computed tomography (CT) scans of the chest, abdomen, and pelvis; whole-body ^18^F-fluorodeoxyglucose (FDG) PET scan; bone marrow aspirate and trephine biopsy; and cerebrospinal fluid (CSF) analysis by cytology. Brain magnetic resonance imaging was recommended in the case of abnormal results in CSF analysis. Patients with baseline CNS involvement were excluded.

The study was approved by the institutional review board of Jeonbuk National University Hospital (document number 2011-03-058) and other each participating institution, and was conducted in accordance with the Declaration of Helsinki. All patients provided written informed consent, and the trial was registered at www.clinicaltrials.gov (NCT01448096). An independent data monitoring committee reviewed the safety and risk/benefit. All authors had access to primary trial data.

### 4.2. Treatments

The patients received six cycles of standard dose of R-CHOP, administered every 3 weeks with the addition of four doses of IT-MTX. R-CHOP consisted of 375 mg/m^2^ rituximab, 750 mg/m^2^ cyclophosphamide, 50 mg/m^2^ doxorubicin, and 1.4 mg/m^2^ vincristine (maximum 2 mg) administered intravenously on day 1 and 100 mg oral prednisone through days 1–5. IT-MTX, 12 mg fixed dose, was administered on day 1 or 2 of each cycle during the first four cycles of treatment. Folinic acid rescue after IT-MTX and infection prophylaxis using trimethoprim-sulfamethoxazole were allowed, as per institutional policy. Prophylactic granulocyte-colony stimulating factor was permitted at the investigator’s discretion. Consolidative radiotherapy for the involved breast and regional nodes was not permitted in patients with CR at the end of R-CHOP, unless they had bulky disease that was defined as any mass with a maximum diameter > 10 cm.

### 4.3. End Points and Assessments

The primary end point of this study was PFS and secondary end points were OS, CNS relapse, and safety. PFS and OS were determined from the date of study enrolment to the date of disease progression, death, or last follow-up, as appropriate. Tumor responses were assessed by local investigators according to the modified International Working Group criteria [31]. Patients underwent response assessment using CT scan after three cycles of systemic therapy (intermediate) and at the end of R-CHOP (final). ^18^F-FDG PET scans were required at baseline and at the end of treatment, but it was recommended (not mandated) after three cycles of R-CHOP as well. Treatment failure was defined as progressive disease on intermediate response assessment and any response less than a partial response on the final assessment. If patients did not have apparent treatment failure on intermediate response assessment, they continued the study treatments and received three additional cycles of R-CHOP. Patients achieving a CR were followed by the treating physicians every 3 months for the first 2 years after treatment and every 4–6 months thereafter.

CNS relapse was diagnosed via magnetic resonance imaging and/or presence of lymphoma cells in the CSF when CNS involvement was clinically suspected. Because the CNS-IPI model was not available during the trial design [32], the CNS-IPI score was retrospectively collected after the completion of patients’ enrolment.

Adverse events were assessed according to the National Cancer Institute Common Terminology Criteria for Adverse Events (version 3.0, Bethesda, MD, USA). Development of subsequent breast cancer was monitored as a specific variable of interest for survivors.

### 4.4. Statistical Analysis

We used a one-side logrank test to calculate the single-arm sample size in this study. In our previous study of patients with primary breast DLBCL who were treated with chemotherapy regimens not including rituximab [7], we reported a 2-year PFS rate of 62%. To show an improvement in the 2-year PFS rate of 23% in patients treated by R-CHOP and IT-MTX combination, we needed 30 patients to be recruited for 7-year accrual and 2-year minimum follow-up (90% power at 0.2 significance level). If we assumed a dropout rate of 10%, the total accrual needed to be 33 patients.

The Kaplan–Meier method was used to estimate PFS and OS, and survival curves were compared using the log-rank test. The cumulative incidence of CNS relapse and subsequent breast cancer was calculated using a cumulative incidence method that incorporated any cause of death as a competing risk. A two-sided *p* < 0.05 was considered significant. All data analysis was carried out using *R* statistical software package (version 3.4.0, Redmond, WA, USA).

## 5. Conclusions

Although the primary end point of PFS was not reached in this study, R-CHOP plus IT-MTX without routine consolidative radiotherapy showed excellent feasibility and resulted in clinically meaningful survival outcomes. However, our study underscores the need for further studies to define risk-stratified novel prophylactic strategies for reducing CNS relapse in patients with primary breast DLBCL. Moreover, our results may provide guidance for designing future trials on novel treatment strategies.

## Figures and Tables

**Figure 1 cancers-12-02192-f001:**
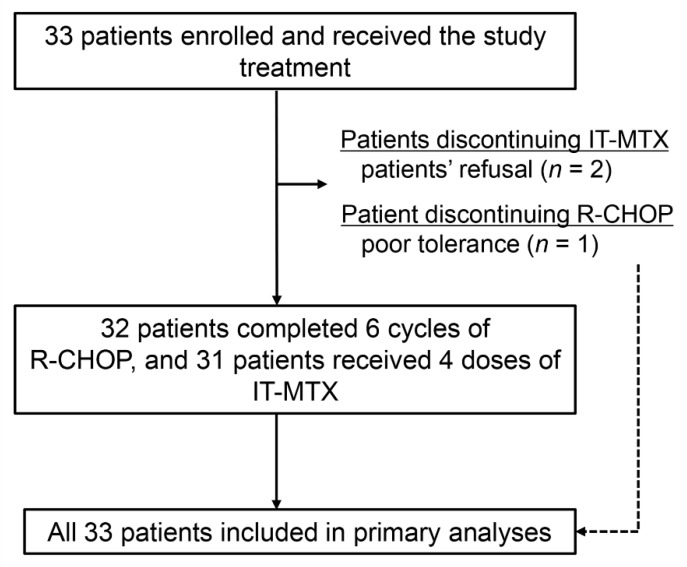
Trial profile. Abbreviations: IT-MTX, intrathecal methotrexate; R-CHOP, rituximab plus cyclophosphamide, doxorubicin, vincristine, and prednisone.

**Figure 2 cancers-12-02192-f002:**
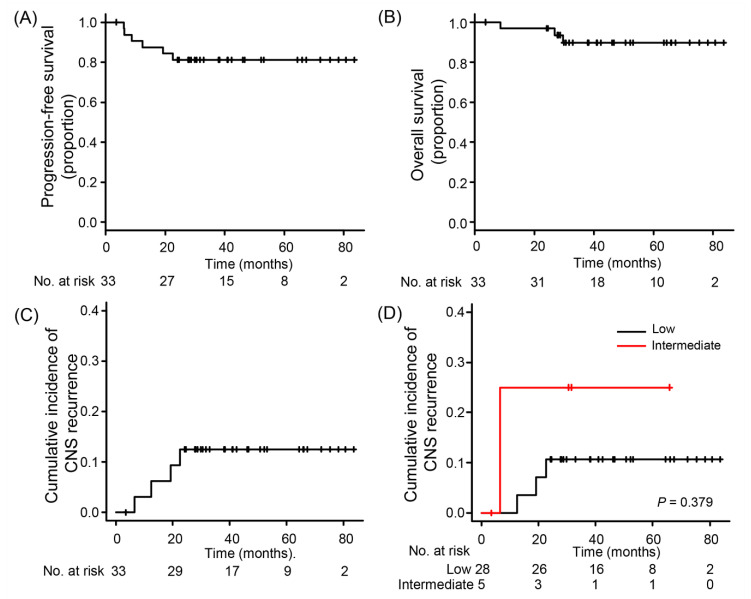
(**A**) Progression-free survival, (**B**) overall survival, (**C**) cumulative incidence of CNS relapse, and (**D**) cumulative incidence of CNS relapse according to CNS-IPI. Abbreviations: CNS, central nervous system; CNS-IPI, central nervous system-International Prognostic Index.

**Table 1 cancers-12-02192-t001:** Baseline clinical characteristics.

Characteristics	Patients (*n* = 33)	
	No.	%
Age, years		
Median	50	
Range	29–75	
Primary site involved		
Right	18	54.5
Left	14	42.4
Bilateral	1	3.0
Lymph node involvement		
No	17	51.5
Regional node	14	42.4
Distant node	2	6.1
Ann Arbor stage		
IE	17	51.5
IIE	13	39.4
IIIE	2	6.1
IVE	1	3.0
ECOG performance status		
0 or 1	32	97.0
2	1	3.0
B symptoms		
Absence	31	93.9
Presence	2	6.1
Serum lactate dehydrogenase level		
Normal	24	72.7
Elevated	9	27.3
Bulky disease		
No	32	97.0
Yes	1	3.0
IPI		
Low	28	84.8
Low-intermediate	2	6.1
High-intermediate	3	9.1
High	0	0
CNS-IPI		
Low	28	84.8
Intermediate	5	15.2
High	0	0

Abbreviations: ECOG, Eastern Cooperative Oncology Group; IPI, International Prognostic Index; CNS-IPI, Central Nervous System-International Prognostic Index.

**Table 2 cancers-12-02192-t002:** Hematologic and non-hematologic toxicity profiles.

Toxicities	Treatment Emergent Adverse Events (Total = 196 Cycles, %)	Grade 1–2		Grade 3–4	
		No.	%	No.	%
Hematological					
Neutropenia	70 (35.7)	2	1.0	68	34.7
Anemia	49 (25.0)	43	21.9	6	3.1
Thrombocytopenia	22 (11.2)	14	7.1	8	4.1
Infectious					
Febrile neutropenia	8 (4.1)	0	0	8	4.1
Other infection	3 (1.5)	2	1.0	1	0.5
Gastrointestinal					
Mucositis	28 (14.3)	26	13.3	2	1.0
Nausea/vomiting	15 (7.7)	14	7.1	1	0.5
Constipation	7 (3.6)	6	3.1	1	0.5
Neurological					
Sensory	66 (33.7)	64	32.7	2	1.0
Motor	10 (5.1)	8	4.1	2	1.0
Central nervous system	1 (0.5)	0	0	1	0.5
Cutaneous	9 (4.6)	9	4.6	0	0
Musculoskeletal	14 (7.1)	14	7.1	0	0

**Table 3 cancers-12-02192-t003:** Subsequent breast cancer in survivors.

Case	Age	Lymphoma Involved	Time From Last Treatment to Breast Cancer Diagnosis (Months)	Radiation Therapy During Lymphoma Treatment	Breast Cancer Developed	Breast Cancer Stage
#5	55	Left	27.0	No	Contralateral	1
#18	47	Left	14.9	No	Ipsilateral	2
#20	46	Right	80.3	No	Ipsilateral	2
#31	50	Right	14.0	No	Ipsilateral	1

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
