# Peer review of "First-Line Treatment for Primary Breast Diffuse Large B-Cell Lymphoma Using Immunochemotherapy and Central Nervous System Prophylaxis: A Multicenter Phase 2 Trial"

_cancers, 2020, doi:10.3390/cancers12082192_

Round 1
Reviewer 1 Report
The manuscript by Yhim et al "First-line treatment for primary breast diffuse large B-cell lymphoma using immunochemotherapy and central nervous system prophylaxis: a multicenter phase 2 trial" is a clearly written concise report of a trial for patients with primary breast DBCL. I find the report compelling and well written.
My only comment is, what was the significance of doing a CT scan after 3 cycles of R-CHOP? was there any strategy and action based on the result of this?
Even though I understand that this is an extremely rare disease, it would be interesting to know what the outcome for this entity has been for other groups, not only their own group, in order to assess the benefit of the intervention.
If CNS relapse is such a threat for these patients, is there a role for CNS prophylaxis with rituximab? or triple IT (MTX, Are-C and hydrocortisone?)
There is no clear definition of CNS involvement, is it malignant cells in CSF? the staging work up did not mention CNS imaging at presentation, how do you know for sure there is no involvement? PET scan may not be the best modality to assess CNS involvement in these patients.
Author Response
Response to Reviewer #1 comments.
The manuscript by Yhim et al "First-line treatment for primary breast diffuse large B-cell lymphoma using immunochemotherapy and central nervous system prophylaxis: a multicenter phase 2 trial" is a clearly written concise report of a trial for patients with primary breast DBCL. I find the report compelling and well written.
- My only comment is, what was the significance of doing a CT scan after 3 cycles of R-CHOP? was there any strategy and action based on the result of this?
: As the reviewer mentioned, routine intermediate (or interim) imaging for DLBCL during immunochemotherapy is controversial and plays a limited role in detecting and predicting treatment failure. Nevertheless, intermediate CT or PET scans for newly diagnosed DLBCL are usually designed in the clinical trials to detect treatment failure during immunochemotherapy, which is generally pursued together with clinical examination and blood tests.
According to such a current status, we assessed the intermediate response after three cycles of R-CHOP and final response after the end of study treatment. In this phase 2 trial, we defined the treatment failure on intermediated and final response assessments as follows; progressive disease on intermediate response assessment and any response less than a partial response on the final assessment. If patients did not have apparent treatment failure on intermediate response assessment, they continued the study treatments and received three additional cycles of R-CHOP.
In a revised version of the manuscript, we stated the response assessment and the strategy based on imaging results in more detail in the materials and methods section (Page 9, Line 261 - 268).
- Even though I understand that this is an extremely rare disease, it would be interesting to know what the outcome for this entity has been for other groups, not only their own group, in order to assess the benefit of the intervention.
If CNS relapse is such a threat for these patients, is there a role for CNS prophylaxis with rituximab? or triple IT (MTX, Are-C and hydrocortisone?)
: The addition of rituximab to systemic chemotherapy has slightly reduced CNS relapse in DLBCL, probably through superior systemic control as there is negligible CNS penetration of the drug across the intact blood-brain barrier. However, recent systematic review for CNS prophylaxis using intrathecal methotrexate and/or cytarabine in patients with DLBCL treated with rituximab or obinutuzumab-based systemic immunochemotherapy revealed no significant reduction of CNS relapse (reference #16 in our report). Instead, reflecting the uncertainty around the efficacy of intrathecal prophylaxis, systemically administered CNS prophylaxis (i.e., high-dose intravenous methotrexate) has been increasingly employed in recent years. Ferreri et al. (reference #17) reported a retrospective analysis of 107 patients with high-risk features for CNS relapse (involvement of specific extranodal sites or advanced stage with high lactate dehydrogenase). In this study, 40 of 107 patients received CNS prophylaxis, the majority receiving high-dose methotrexate ± intrathecal therapy. The CNS relapse rate in patients who received prophylaxis was 2.5% compared to 12% in those who did not, although the number of patients with high CNS-IPI was lower in the prophylaxis group. Besides, it is noteworthy that a patient with CNS relapse in the prophylaxis group received only intrathecal methotrexate without systemic prophylaxis, suggesting that intrathecal prophylaxis alone might be inadequate for preventing CNS relapse in high-risk DLBCL patients. However, because there has been no randomized study demonstrating a benefit of high-dose methotrexate CNS prophylaxis, there remains a lack of consensus regarding appropriate timing, dose, and number of cycles. In our study group (Consortium for Improving Survival of Lymphoma; CISL), we are now conducting a randomized phase 3 clinical trial (PICASSO trial, NCT03123718) comparing high-dose intravenous methotrexate (3.0 g/m2; N=103) after second and sixth cycles of R-CHOP with four doses of intrathecal methotrexate (15mg; N=102) in newly diagnosed DLBCL patients with high risk for CNS relapse. We expect that patient’s enrollment will be completed at the end of this year.
In a revised version of the manuscript, we stated it in more detail in the discussion section (Page 6, Line 148 - Page 7, Line 150) and added the references (#15 and #16).
- There is no clear definition of CNS involvement, is it malignant cells in CSF? the staging work up did not mention CNS imaging at presentation, how do you know for sure there is no involvement? PET scan may not be the best modality to assess CNS involvement in these patients
: All patients included in this trial underwent cerebrospinal fluid (CSF) analysis with cytologic evaluation before study enrollment. If patients would have abnormal CSF results such as positive lymphoma cells and/or pleocytosis, they were recommended to perform brain MRI. CNS involvement was defined as the presence of lymphoma cells in the CSF and/or lymphoma lesions on brain MRI. Patients were finally excluded if they were diagnosed with concurrent CNS involvement. As we already stated in the results section (Page 2, Line 79 - 80), none of the patients enrolled in this trial had baseline CNS involvement.
In a revised version of the manuscript, we added it in the materials and methods section (Page 8, Line 238 - 239).
Reviewer 2 Report
The authors present a prospective multicenter phase 2 study of R-CHOP with IT methotrexate for primary breast DLBCL. With a median follow-up of 46.1 months, the 2-year PFS and OS were 81.3% and 93.5%, respectively with 6 relapses (4 in CNS and 2 in the breast).
Overall the manuscript is well written. Given the rarity of the disease, a prospective phase 3 study in this patient population will likely not be conducted, hence, the data presented here is important. The major limitation is the small sample size which the authors’ have rightfully outlined in the limitations section. I have a few comments for the authors’ consideration.
Comments:
- How did you define bulky in the study?
- Please provide the COO in the baseline characteristics Table (GCB vs non-GCB)
- Please provide the number of patients who harbored c-MYC/BCL2 expression status by IHC (double expressors) and by FISH (double/triple hit) in the study. This is important.
- Despite approx. half of the study population being stage IE, it is disheartening to note the high rates of relapse (esp CNS relapse). Did any of these stage IE patients relapse?
- Please also provide an additional discussion on the recent cluster classification and how it can have an impact on treatment selection
- A minor point, please have the materials and methods section before the results section for a better flow of the manuscript.
Author Response
Response to Reviewer #2 comments.
The authors present a prospective multicenter phase 2 study of R-CHOP with IT methotrexate for primary breast DLBCL. With a median follow-up of 46.1 months, the 2-year PFS and OS were 81.3% and 93.5%, respectively with 6 relapses (4 in CNS and 2 in the breast).
Overall the manuscript is well written. Given the rarity of the disease, a prospective phase 3 study in this patient population will likely not be conducted, hence, the data presented here is important. The major limitation is the small sample size which the authors’ have rightfully outlined in the limitations section. I have a few comments for the authors’ consideration.
Comments:
- How did you define bulky in the study?
: Bulky disease was defined as any mass with a maximum diameter > 10cm, which was described in the original version of our manuscript (Page 9, Line 255 - 256).
2 & 3. Please provide the COO in the baseline characteristics Table (GCB vs non-GCB).
Please provide the number of patients who harbored c-MYC/BCL2 expression status by IHC (double expressors) and by FISH (double/triple hit) in the study. This is important.
: We agree with the reviewer’s comments that COO, double expressor, and double-hit/triple-hit (DH/TH) status on tumor tissue may provide crucial information on CNS outcomes in patients with primary breast DLBCL. However, because it had not been fully established at the time of the trial design, archiving tumor tissue was not initially planned for this study. Because of this, unfortunately, we cannot provide the data for COO, double expressor, and DH/TH status in patients enrolled in this trial.
However, we are now planning to conduct a biological correlative study using tumor tissues of patients with primary breast DLBCL, in which we are going to evaluate the impact of COO, double expressor, and DH/TH status on CNS outcomes. The study population consists of two primary breast DLBCL cohorts of our study group (one retrospective cohort and the present cohort). The retrospective cohort of patients with primary breast DLBCL was previously published (Yhim HY et al. Int J Cancer 2012;131:235-43).
Therefore, we do not provide the data for COO, double expressor, and DH/TH status that the reviewer asked, but we will be able to address the reviewer’s questions in another study that is now underway. We ask for the reviewer’s understanding of this.
- Despite approx. half of the study population being stage IE, it is disheartening to note the high rates of relapse (esp CNS relapse). Did any of these stage IE patients relapse?
: Among 17 patients (51.5%) with stage IE disease, CNS relapse occurred in two patients at 12.4 and 19.2 months, respectively. Therefore, the 2-year cumulative incidence of CNS relapse of patients with stage IE disease was 11.7% (95% CI, 0-25.8), which was not statistically different from that in patients with stage IIE, IIIE, or IV (13.3%; 95% CI, 0-28.9; p = 0.896). We also provided a compatible cumulative incidence plot as follows.
In addition to the Ann Arbor stage, we also evaluated the association between other baseline clinical variables (i.e., age, B symptoms, LDH level, bulky disease status and so on) and CNS relapse and found that none of the clinical variables were significantly associated with CNS relapse (Table S1.).
In a revised version of the manuscript, we briefly stated it in the results section (Page 5, Line 116 - 117) and supplementary materials (Table S1.).
- Please also provide an additional discussion on the recent cluster classification and how it can have an impact on treatment selection
: In an original version of the manuscript, we briefly described the needs of investigation about COO, double expressor, and DH/TH status of the tumors to account for peculiar clinical features of primary breast DLBCL (Page 7, Line 160 - 171). In a revised version of the manuscript, we added the statement that a molecularly defined high-grade cluster has recently been identified. This subgroup was a clinically and biologically distinct cluster of DLBCL that was characterized by a similar gene expression signature of DH/TH lymphoma. Also, we stated a potential impact of genetic classifications on CNS prophylactic strategy in patients with primary breast DLBCL (Page 7, Line 164 - 166; Page 7, Line 172 - 181).
- A minor point, please have the materials and methods section before the results section for a better flow of the manuscript.
: Manuscript format and sequence that the Cancers require is Introduction, Results, Discussion, Materials and Methods, and Conclusions. Please, make sure that this is not what we can handle.

Reviewer 3 Report
Dr Yhim and collaborators have described the results of a phase 2 trial testing the usefulness of immunochemotherapy and intrathecal prophylaxis with methotrexate in primary breast DLBCL. Overall, the manuscript is well presented and well written.
Beyond the interest of the results regarding PFS, this manuscript presents interesting data for potential Cancers readers in relation to CNS outcome.
Major comments:
Several recent studies have proposed that the number of cycles of R-CHOP to be given in low risk DLBCL can be reduced to four. Since this study includes 84.8% of low risk IPI patients, a comment might addressing this point in the discussion would be of interest.
Did the 2 patients who did not received full IT-MTX suffer CNS relapse?.
In the original CNS-IPI, breast origin of the DLBCL did not reach statistical significance in the univariate analysis. However, the number of patients with breast DLBCL was not described in that study. I will emphasize in the discussion that, considering previous retrospective data and present results, patients with breast DLBCL should be considered for more active prophylactic strategies. In this point, new small molecules such as ibrutinib and lenalidomide might be considered to be tested in this specific setting. Some comments regarding this issue might be also of interest for the readers.
I believe that PFS and OS results are pretty good in this small and probably selected population of patients (as usual in phase 2 trials). However, I understand that the main result of this trial is that IT-MTX is not an optimal strategy for prevention of CNS disease in breast DLBCL (even with low CNS-IPI) and further studies are needed. Regarding this issue, in the abstract, the following sentence might induce to misunderstanding: “Standard R-CHOP and IT-MTX without routine radiotherapy offers clinically meaningful survival outcomes in patients with primary breast DLBCL”. I suggest to rephrase this sentence.
Author Response
Response to Reviewer #3 comments.
Dr Yhim and collaborators have described the results of a phase 2 trial testing the usefulness of immunochemotherapy and intrathecal prophylaxis with methotrexate in primary breast DLBCL. Overall, the manuscript is well presented and well written.
Beyond the interest of the results regarding PFS, this manuscript presents interesting data for potential Cancers readers in relation to CNS outcome.
Major comments:
- Several recent studies have proposed that the number of cycles of R-CHOP to be given in low risk DLBCL can be reduced to four. Since this study includes 84.8% of low risk IPI patients, a comment might addressing this point in the discussion would be of interest.
: As the reviewer mentioned, some patients with primary breast DLBCL might be treated with short-course immunochemotherapy without radiotherapy; however, we need to consider that approximately 20% of the patients would have systemic failures (i.e., CNS and contralateral breast) after full-course immunochemotherapy. In this context, we think that such an approach should be compared after the impact of biologic prognostic factors such as cell-of-origin, double expressor, and double-hit/triple-hit status on clinical outcomes was completely defined.
In a revised version of the manuscript, we added the statement regarding risk-based treatment de-escalation in primary breast DLBCL patients without adverse biologic features (Page 7, Line 172 - 178).
- Did the 2 patients who did not received full IT-MTX suffer CNS relapse?.
: As we already described in the original version of the manuscript (Page 5, Line 111 - 112), all four patients with CNS relapses had completed full study treatments (i.e., six cycles of R-CHOP and four doses of IT-MTX). Treatment failure was not observed in two patients received only two and three doses of IT-MTX.
- In the original CNS-IPI, breast origin of the DLBCL did not reach statistical significance in the univariate analysis. However, the number of patients with breast DLBCL was not described in that study. I will emphasize in the discussion that, considering previous retrospective data and present results, patients with breast DLBCL should be considered for more active prophylactic strategies. In this point, new small molecules such as ibrutinib and lenalidomide might be considered to be tested in this specific setting. Some comments regarding this issue might be also of interest for the readers.
: In a revised version of the manuscript, we added the statements regarding the potential role of ibrutinib and lenalidomide in primary breast DLBCL with adverse biologic features for CNS failure (Page 7, Line 179 - 184).
- I believe that PFS and OS results are pretty good in this small and probably selected population of patients (as usual in phase 2 trials). However, I understand that the main result of this trial is that IT-MTX is not an optimal strategy for prevention of CNS disease in breast DLBCL (even with low CNS-IPI) and further studies are needed. Regarding this issue, in the abstract, the following sentence might induce to misunderstanding: “Standard R-CHOP and IT-MTX without routine radiotherapy offers clinically meaningful survival outcomes in patients with primary breast DLBCL”. I suggest to rephrase this sentence.
: In a revised version of the manuscript, we revised the sentence that the reviewer recommended to rephrase (Page 1, Line 34 - 36).